# The Impact of a Woodland Walk on Body Image: A Field Experiment and an Assessment of Dispositional and Environmental Determinants

**DOI:** 10.3390/ijerph192114548

**Published:** 2022-11-05

**Authors:** Kamila Czepczor-Bernat, Justyna Modrzejewska, Adriana Modrzejewska, Viren Swami

**Affiliations:** 1Department of Pediatrics, Pediatric Obesity and Metabolic Bone Diseases, Faculty of Medical Sciences in Katowice, Medical University of Silesia, 40-055 Katowice, Poland; 2Institute of Pedagogy, University of Bielsko-Biała, 43-309 Bielsko-Biala, Poland; 3Department of Psychology, Chair of Social Sciences and Humanities, School of Health Sciences in Katowice, Medical University of Silesia, 40-055 Katowice, Poland; 4School of Psychology and Sport Science, Anglia Ruskin University, Cambridge, Cambridgeshire CB1 1PT, UK; 5Centre for Psychological Medicine, Perdana University, Kuala Lumpur 50490, Malaysia

**Keywords:** nature exposure, body appreciation, positive body image, field experiment, self-compassion

## Abstract

Studies have shown that nature exposure is associated with a more positive body image, but field studies remain relatively infrequent. Here, we examine the impact of a woodland walk on an index of state positive body image (i.e., state body appreciation), as well as dispositional and environmental determinants of body image improvements. Eighty-seven Polish women went for a walk in Cygański Las, an ancient woodland, and completed a measure of state body appreciation before and after the walk. As hypothesised, state body appreciation was significantly higher post-walk compared to pre-walk (*d* = 0.56). Additionally, we found that the trait of self-compassion—but not the traits of connectedness to nature, perceived aesthetic qualities of the woodland, or subjective restoration—was significantly associated with larger improvements in state body appreciation. These results suggest that even relatively brief exposure to nature results in elevated state body appreciation, with the dispositional trait of self-compassion being associated with larger effects.

## 1. Introduction

A large body of evidence now exists showing that nature exposure—living close to, frequenting, or engaging with natural environments, such as forests and urban parks—is associated with a range of benefits for physical and psychological well-being [1,2,3]. These effects include positive psychological functioning [4], which also involves improvements to body image. In particular, a growing body of evidence suggests that nature exposure is reliably associated with multiple indices of positive body image, which Tylka [5] defined as an “overarching love and respect for the body” (p. 9) that includes appreciation of the body and its functions, acceptance of the body despite its imperfections, and body-protective behaviours. The promotion of a positive body image is important not only in its own right [6], but also because of its beneficial downstream effects, including more positive psychological well-being, adaptive eating styles that are associated with weight stability, and flourishing [7,8,9].

The evidence base supporting an association between nature and a positive body image has three main sources. First, cross-sectional studies [10,11,12,13,14] and an experience sampling study [15] have shown that self-reported nature exposure is significantly associated with multiple indices of the traits of a positive body image, including body appreciation and functionality appreciation. Second, experimental studies have shown that exposure to both real and simulated (i.e., images or film) natural environments results in an elevated state body appreciation [11,16,17,18]. A third class of studies offers more direct evidence of the impact of nature exposure on state body image in everyday settings: single-arm pre-post studies have shown that spending time in natural environments (e.g., gardening on an allotment, going for a hike, walking in botanic gardens) significantly elevates state body appreciation [11,19,20].

The effects of nature exposure on positive body image have mainly been explained by drawing on attention restoration theory [21,22]. From this perspective [23], it is proposed that individuals benefit mentally from the opportunities provided by natural environments to “be away”, experience expansive spaces and contexts (“extent”), engage in activities that are “compatible” with intrinsic motivations, and critically experience stimuli that are “softly fascinating”. These characteristics of natural environments are thought to restrict negative appearance-related thoughts [11,24] and to shift attention away from an aesthetic view of the body and toward greater appreciation of the body’s functionality [12]. Additionally, nature exposure may also help to promote holistic self-care attitudes and behaviours—such as positive coping strategies—that result in greater respect, appreciation, and love for one’s body [25].

Despite the wealth of theorising and empirical evidence, little is currently known about the determinants of positive body image outcomes in everyday natural environments. This is important because, to the extent that natural environments offer a cost-effective and readily available method of promoting positive body image [24], it will be necessary for understanding mechanistic pathways more deeply. That is, although the impact of natural environments on positive body image is now well-documented, more can be completed to understand the specific determinants of such effects in everyday encounters and experiences, which in turn could assist in the development of more effective interventionist strategies. To wit, it will be important to consider the impact of both dispositional traits (i.e., person-centred factors) and environmental factors (i.e., features of the environment in which nature exposure occurs or an individual’s perceptions and understandings of those features) [26].

### 1.1. Dispositional Traits

Although it is generally agreed that natural environments promote improved psychological well-being [1,2,3], it is also assumed that any benefit is the product of a person–environment interaction [27]. For instance, place identity theory [28] suggests firstly that attitudinal dispositions, preferences, and memories of physical environments help to shape self-identities; it also hypothesises that place identity varies as a function of demographic characteristics (e.g., gender, social class), personality, and other dispositional traits. These factors, in turn, may affect one’s experiences in natural environments and therefore potentially shape the magnitude of the positive effects accrued by being in nature [27]. In support of this general account, studies have suggested that the outcomes of nature exposure (e.g., restoration, psychological well-being) are affected by various dispositional and individual difference traits, such as urban–nature orientedness, noise sensitivity, and need for restoration [27,29].

In terms of body image outcomes, two primary dispositional traits have been highlighted as particularly relevant [30], namely connectedness to nature and self-compassion. In the first instance, connectedness to nature—which refers to a sense of oneness with nature [31]—has been found to mediate the relationship between nature exposure and positive body image [32]. As an explanation, it has been suggested that greater or repeated nature exposure promotes greater connectedness to nature [33], which in turn facilitates perceptions of the self as requiring care within broader ecological systems [32]. In the second instance, self-compassion (i.e., an ability to be kind to oneself) [34] has likewise been shown to mediate the relationship between nature exposure and positive body image [12,13], possibly by facilitating a recognition that everyone has imperfections and by encouraging individuals to show kindness and acceptance towards their physical selves.

Additionally, both connectedness to nature and self-compassion may also promote more embodying experiences in natural environments. Embodiment theory suggests that pleasurable engagement in joyful physical activity should enhance positive connections with the body in both the short- and long-term [35,36]. That is, individuals with a sense of positive embodiment may be more likely to experience their bodies in positive ways (e.g., by focusing on and respecting their bodies’ functional qualities [35]), which in turn may be expected to improve state body image experiences, as well as trait body image, in the longer term. To date, however, the evidence base linking nature exposure to positive body image via connectedness to nature and self-compassion, respectively, remains limited to cross-sectional work. That is, little is known about the possible ways in which these dispositional traits affect body image outcomes in more naturalistic, everyday settings.

### 1.2. Environmental Factors

Beyond dispositional traits, perceptions of and experiences in natural environments are also known to affect the outcomes of nature exposure. For instance, drawing on attention restoration theory [21,22], it has been suggested that natural environments are most beneficial when they are experienced as being restorative [26]. In this view, restoration is defined as a short-term, mood-like state involving affective, physiological, and cognitive recovery [21]. A restorative environment, therefore, is one that is experienced as promoting recovery and positive responses [37]. As such, it can be expected that individuals who experience a natural environment as being more restorative will also experience greater improvements in short-term nature-related outcomes [26], although this has only been evidenced cross-sectionally in relation to body image outcomes [10].

Additionally, perceptions of the aesthetic qualities of an environment may also affect the outcomes of nature exposure [38,39]. In Ulrich’s [40] stress reduction theory, for instance, it is assumed that the aesthetic and visual perception of an environment triggers immediate and unconscious affective responses, such as preference and liking, which in turn can influence the outcomes of nature exposure. In support of this general perspective, studies have suggested that perceptions of the qualities of an environment—such as perceived biodiversity, naturalness, and visual appeal—are associated with greater restoration and psychological well-being [27,41,42]. In terms of body image outcomes specifically, some research has postulated that environmental factors—such as perceived cleanliness and biodiversity—may affect outcomes [20], but this has not been tested empirically.

### 1.3. The Present Study

The review above suggests that much more can be undertaken to better understand dispositional and environment-related factors that may affect body image outcomes in natural environments. To that end, we report on a field study designed to test some of the questions raised herein. First, utilising a pre-post study design, we sought to examine the impact of a walk in a natural environment (an ancient woodland) on state body appreciation. Unlike previous work [11,19,20], where participants have been tested individually during spring or summer months, the present study tested participants in a group setting during winter months. Here, we predicted that walking in the ancient woodland would significantly elevate state body appreciation, which would be consistent with previous work [11,19,20].

Additionally, we also assessed the extent to which two dispositional traits—connectedness to nature and self-compassion—affected the magnitude of state body image improvements as a result of the walk. We hypothesised that both greater connectedness to nature and self-compassion would be significantly associated with larger improvements in state body image. In terms of environmental factors, we considered the extent to which perceived environmental aesthetic qualities and perceived restoration in the natural setting would be associated with changes in state body image. Our expectation in this regard was that more positive aesthetic evaluations of the natural environment and greater perceived restoration in the natural environment would be significantly associated with larger improvements in state body image.

## 2. Materials and Methods

### 2.1. Participants

An a priori power analysis based on Study 4 in [11] that assumed a fully within-subjects design indicated that a minimum sample size of 76 was needed. Initially, 161 participants enrolled in the study and completed all baseline measures. However, only 91 participants (87 women and 4 men) completed the field phase of the study involving the walk in the natural environment. Due to the small number of men, they were excluded from analyses, leaving a sample of 87 women. This final sample size exceeded the initial requirement based on the power calculation. The sample of women ranged in age from 19 to 55 years (*M* = 23.85, *SD* = 5.23) and in self-reported body mass index (BMI) from 16.93 to 33.61 kg/m^2^ (*M* = 22.29, *SD* = 3.34). The majority of participants were white (97.7%). In terms of education, 57.5% had completed secondary or technical school, 36.8% had completed an undergraduate degree, 2.3% had completed a Master’s degree, and 3.5% had completed some other qualification.

### 2.2. Measures

#### 2.2.1. Baseline Measures

At baseline (3–4 weeks before the experimental phase; see Section 2.3), participants were asked to provide their demographic details (gender identity, age, highest educational qualifications, race, weight, and height) and complete two dispositional measures.

#### 2.2.2. Self-Compassion

To measure the trait of self-compassion, participants were asked to complete a Polish translation [43] of the 12-item Short Form of the Self-Compassion Scale (SCS-SF) [44], which measures aspects of self-kindness, common humanity, and mindfulness as defined in Neff’s [45] model of self-compassion. All items were rated on a 5-point scale, ranging from 1 (*almost never*) to 5 (*almost always*). Scores on the Polish version of the SCS-SF were shown to be unidimensional with adequate construct validity [43]. An overall score was computed as the mean of all items following the reverse coding of six items, with higher scores reflecting greater self-compassion. The internal consistency, as measured using McDonald’s ω, for SCS-SF scores in the present study was 0.92 (95% CI = 0.90, 0.94).

#### 2.2.3. Connectedness to Nature

To measure participants’ perceived oneness with nature, we used the Connectedness to Nature Scale (CNS) [31]. The CNS consists of 14 items that are rated on a five-point scale ranging from 1 (*strongly disagree*) to 5 (*strongly agree*). A Polish version of the instrument was previously translated for use in the Body Image in Nature Survey [30], but we are not aware of any previous assessment of its factorial validity. We, therefore, subjected our data to a principal–axis exploratory factor analysis (EFA), the results of which supported the retention of a single factor consisting of 9 of the 14 items (KMO = 0.90, Bartlett’s test of sphericity, *χ*^2^(36) = 558.53, *p* < 0.001, eigenvalue = 5.79, 64.30% of the variance explained, item-factor loadings = 0.74 to 0.87). The removal of several items (# 4, 6, 12, 13, 14 in the present study) is consistent with outcomes of factor analyses in other, non-English-speaking national contexts [46]. The internal consistency for the 9-item CNS used in the present study was 0.93 (95% CI = 0.91, 0.95).

#### 2.2.4. Field Measures

During the field phase of the study, the participants were asked to complete the following measures.

#### 2.2.5. State Body Appreciation

Pre- and post-nature exposure, participants were asked to complete a state version of the 10-item Body Appreciation Scale-2 (SBAS-2) [47]. In this version of the BAS-2, items are worded to reflect time-specific states of positive body image. All items were rated on a five-point scale ranging from 1 (*strongly disagree*) to 5 (*strongly agree*). Due to the fact that this specific measure has not been previously used in Polish, we first adapted items from the Polish version of the BAS-2 [48] to reflect the state version, as per [47]. Next, we subjected pre- and post-nature exposure data to principal-axis EFAs, which supported the extraction of a single factor consisting of all 10 items (pre-exposure: KMO = 0.93, Bartlett’s test of sphericity, *χ*^2^(45) = 866.34, *p* < 0.001, eigenvalue = 7.26, 72.55% of the variance explained, item-factor loadings = 0.62 to 0.94; post-exposure: KMO = 0.93, Bartlett’s test of sphericity, *χ*^2^(45) = 1026.41, *p* < 0.001, eigenvalue = 7.87, 78.72% of the variance explained, item-factor loadings = 0.84 to 0.92). The McDonald’s ω for SBAS-2 scores at pre-exposure was 0.96 (95% CI = 0.94, 0.97) and at post-exposure was 0.97 (95% CI = 0.96, 0.98).

#### 2.2.6. Perceived Environmental Aesthetic Qualities

To measure participants’ perceptions of the aesthetic qualities of our field site, we used the Perceived Environmental Aesthetic Qualities Scale (PEAQS) [39]. This is a 23-item instrument that measures perceptions of a physical space along five dimensions, namely Harmony (eight items assessing the degree to which a space reflects balance, unity, and legibility), Mystery (five items that assess the degree to which a space is complex and generates feelings of excitement and desire for exploration), Multisensority and Nature (four items assessing the degree of diversity in sensory inputs in a space), Visual Spaciousness and Visual Diversity (three items assessing the degree of visual diversity and perceived spaciousness), and Sublimity (three items assessing the degree to which a space triggers feelings of awe). All items were rated on a seven-point scale ranging from 1 (not at all) to 7 (completely). A Polish translation of the PEAQS was prepared for the present study following the test adaptation recommendations in [49]. When we subjected our data to principal–axis EFA, we found that all 23 items loaded onto a single factor (KMO = 0.94, Bartlett’s test of sphericity, χ^2^(253) = 2768.85, *p* < 0.001, eigenvalue = 16.01, 69.62% of the variance explained, item-factor loadings = 0.63 to 0.92). We, therefore, computed an overall score as the mean of all 23 items, with higher scores reflecting more positive perceptions of the aesthetics qualities of the field site. The internal consistency for this overall score was 0.98 (95% CI = 0.97, 0.99).

#### 2.2.7. Restoration

To measure subjective restoration as a result of exposure to the field site, participants were asked to complete the Restoration Outcome Scale (ROS) [50,51]. This is a nine-item instrument that measures the degree of restorative outcomes in terms of relaxation, calmness, attention restoration, clarity of thought, subjective vitality, and self-confidence. All items were rated on a seven-point scale ranging from 1 (not at all) to 7 (completely). A Polish version of the ROS was obtained from the Body Image in Nature Survey [30], and data from the present study were subjected to a principal–axis EFA. The results support the extraction of a single factor consisting of all nine items (KMO = 0.93, Bartlett’s test of sphericity χ^2^(36) = 1101.59, *p* < 0.001, eigenvalue = 7.52, 83.56% of the variance explained, item-factor loadings = 0.88 to 0.94). The McDonald’s ω for ROS scores in the present study was 0.98 (95% CI = 0.97, 0.98).

### 2.3. Procedures

Ethics approval was obtained from the relevant departmental Ethics Committee at the University of Bielsko-Biała (no. 2021/11/7E/8). All participants were university students enrolled in a Pedagogy course. As part of the course, students were informed in November 2021 about the possibility of participating in the project (see Figure 1). In order to mask the main study hypothesis, the project was advertised as a study of the effects of personality on greenspace use. Participants who agreed to participate in the study were sent a link containing brief information about the study and a request for informed consent. At this point, participants who agreed to participate were asked to provide their demographic details and complete the baseline measures (SCS-SF and CNS), along with a Polish translation [52] of the Ten-Item Personality Inventory [53], which we used to mask the study hypotheses.

Three to four weeks after baseline testing, in November–December 2021, the experimental phase of the project took place at Cygański Las, an ancient woodland in the city of Bielsko-Biała. During this testing period, Cygański Las was snow covered. On fair days, between 8 a.m. and 4 p.m., participants were accompanied to Cygański Las. Immediately before entering the woodland, participants completed the SBAS-2 on a mobile device, with surveys presented using Google Forms. Next, in groups of about 15, they went for a single walk in the woodlands for about 40 min on average. Participants were not given any explicit instructions about how to behave during the walk, except to behave naturally as if on an everyday walk, and participants were allowed to interact with each other during the walk. Participants walked one of the paths in the woodlands and were accompanied by one of the researchers who guided the group along the path. At the end of the walk, participants were asked to complete the SBAS-2 again, alongside the PEAQS and the ROS. At the end of testing, participants were fully debriefed. Each student was assigned a unique researcher-generated ID to link their baseline, pre-test, and post-test data. Participants were not remunerated, and all participation was voluntary.

### 2.4. Statistical Analyses

IBM SPSS Statistic v.26 was used to conduct our analyses. To test the main hypothesis (change in SBAS-2 scores between pre- and post-walk), we computed a paired-samples *t*-test with dependence-corrected effect sizes [54]. Before conducting this analysis, we checked the normality of the distribution of both measurements of the SBAS-2. Normality was not met at both time-points (pre-exposure: *W*(87) = 0.95, *p* < 0.001; post-exposure *W*(87) = 0.91, *p* < 0.001), though skewness and kurtosis were acceptable (pre-exposure skewness = −0.78, kurtosis = 0.77; post-exposure skewness = −1.05, kurtosis = 1.70). However, analyses with the non-parametric Wilcoxon Signed Ranks test provided similar results to the paired-samples *t*-test, so we report on the latter here.

For further analyses, we first computed a state body image change score by taking the difference between SBAS-2 scores at pre- and post-exposure. Next, we computed Pearson’s correlations between this score and scores on the CNS, SCS-SF, PEAQS, and ROS tests, respectively. Finally, we computed a hierarchical regression with the state body image change scores as the criterion variable. In the first step, we entered the dispositional traits measured by the CNS and SCS-SF. In the second step, we entered the PEAQS subscale scores and the ROS. All assumptions for the multiple regression analysis were met. Multicollinearity was measured using variance inflation factors (VIFs) and tolerance. All VIFs were < 2.0, indicative of a lack of multicollinearity [55].

## 3. Results

### 3.1. Main Analysis

A paired-samples *t*-test indicated that the state body appreciation scores post-walk (*M* = 3.90, *SD* = 0.88) were significantly higher than those pre-walk (*M* = 3.61, *SD* = 0.92), *t*(86) = 4.80, *p* < 0.001, *d* = 0.56, which supports our primary hypothesis.

### 3.2. Further Analysis

Pearson’s correlations between the body image change scores and scores on the CNS, SCS-SF, PEAQS, and ROS are reported in Table 1. As can be seen, a higher body appreciation change was only significantly and positively associated with self-compassion. Notably, other associations were in the expected directions, including the positive and strong relationship between the perceived aesthetic qualities of the field site and restoration. Next, we conducted a hierarchical multiple regression with body image change scores as the criterion variable (see Table 2). The first step of the regression was significant, *F*(2, 84) = 4.63, *p* = 0.012, Adj. *R*^2^ = 0.08, with both self-compassion and connectedness to nature emerging as significant predictors. The second step of the regression was also significant, *F*(4, 82) = 3.24, *p* = 0.016, Adj. *R*^2^ = 0.09, though Δ*F* was not significant, Δ*F*(2, 82) = 1.76, *p* = 0.177. Self-compassion was the only significant predictor in the second step of the regression.

## 4. Discussion

In the present study, we examined the impact of a walk in an ancient woodland on state body appreciation outcomes in a sample of Polish women. Our results confirm our hypothesis that going for a walk in a natural environment significantly elevates state body appreciation scores. Overall, this finding is consistent with previous work showing that time spent on an allotment [19], in a designed greenspace [11], at the beach, and in botanic gardens [20] significantly elevated state body appreciation scores in populations from diverse national settings. Indeed, the magnitude of the effect was comparable in effect size to some earlier studies [11,13]. However, in contrast to previous work, in which participants were tested individually and during spring or summer months, our results are the first to indicate that the positive effects of spending time in a natural environment on state body appreciation also occur in group settings and during the winter months (and, more precisely, when our field site was snow covered).

It is possible to explain these findings by drawing on attention restoration theory [21,22], which suggests that natural environments have the capacity to restore depleted psychological resources. More specifically, it has been suggested that natural environments may offer opportunities to promote positive body image by restricting negative appearance-related thoughts and supporting speedier recovery from threats to body image, thus turning negative body image states into positive ones [11,24]. Given that participants in the present study were engaged in physical activity (i.e., walking), this may have also helped to shift attention from an aesthetic view of the body to a greater appreciation of the body’s functions [12]. That is, through engagement in a form of physical activity in a restorative natural environment, participants may have come to focus more explicitly on a sense of gratitude for what their bodies allowed them to accomplish. Additionally, the restorative setting of the ancient woodland in the present study may also have facilitated self-care attitudes (e.g., self-compassion) that resulted in greater appreciation for one’s body [25].

Our finding that nature exposure elevated state body appreciation in a snow-covered setting is also noteworthy. That is, where previous studies have focused on the impact of blue and green natural environments on body image outcomes, ours is the first to suggest that white natural environments may also have a similar effect. From a broad perspective, this finding is consistent with previous work suggesting that exposure to white natural environments is associated with greater emotional well-being [56,57]. Of course, wintry conditions are likely to affect well-being outcomes in complex ways. For instance, winter may negatively affect well-being by limiting the availability of pleasant outdoor experiences and through decreased comfort due to cold temperatures [58,59]. In future research, it may be worth further interrogating this aspect of our findings. For instance, it might be worth examining the moderating role of a positive wintertime mindset on body image outcomes in white natural environments [60].

Additionally, and in an extension of existing knowledge, we also found that the trait of self-compassion was significantly associated with a greater magnitude of change in state body appreciation. This is broadly consistent with previous cross-sectional work showing that self-compassion significantly mediated the relationship between nature exposure and body appreciation [12]. It is likely that individuals who are high in the trait of self-compassion have dispositional characteristics that allow them to maximally benefit from nature exposure. For instance, the deliberation without attention that occurs in natural environments [61,62] may allow individuals who are high in self-compassion to calm their minds [63] or reach a state of relaxation more quickly [64], which in turn may lead to larger effects on state body image. In particular, it is possible that being in nature facilitates recognition of the fact that everyone has imperfections and encourages individuals to show kindness and acceptance towards their bodies [65,66]—aptitudes that may occur more quickly or strongly in individuals who are high in self-compassion.

In contrast, the dispositional trait of connectedness to nature did not emerge as a significant predictor of state body appreciation change once the effects of environmental-related factors had been taken into account (though it was a significant predictor when included in isolation with self-compassion). The most likely explanation for this effect is that connectedness to nature is only weakly associated with body image outcomes in natural environments. For example, although previous work has shown that connectedness to nature mediates the relationship between nature exposure and body appreciation [13], the direct relationship between connectedness to nature and body appreciation was weak. It is also possible that this result was affected by the fact that we used a truncated version of the CNS, based on the results of our factor analysis. Although the need to eliminate items to achieve an adequate unidimensional fit is consistent with previous work [46], it is possible that truncating the CNS resulted in the loss of conceptual meaning, which affected our findings.

Perhaps more interestingly, we found that neither the perceived aesthetic qualities of the field site nor subjective restoration were significantly associated with changes to state body appreciation. In the first instance, this stands in contrast to studies showing that perceptions of the qualities of an environment are associated with greater restoration and psychological well-being [27,41,42]. In the second instance, the null effect *vis-à-vis* subjective restoration stands in contrast to the predictions of attention restoration theory [21,22] as well as cross-sectional work showing that recalled restoration is significantly associated with body appreciation [10]. One explanation for the present findings is that environmental factors may not exert much of an effect on state body appreciation changes in natural environments once the effects of dispositional traits have been accounted for. That is, from a holistic perspective, it may be that dispositional traits trump environmental factors in terms of affecting state body appreciation changes.

An alternative, though not mutually exclusive, explanation is there were floor effects in our PEAQS and ROS scores. That is, there may have been limited variance in scores on these measures, which created a non-extendable “floor” [67] and, in turn, dampened any association with state body appreciation change. It is also of note that, in the present study, we computed an overall score for the PEAQS, which was consistent with the results of our factor analysis. However, this meant that we were unable to assess associations with specific perceptions of the environment, as measured in the original form of the PEAQS (e.g., perceptions of spaciousness and diversity, harmony, and so on). Thus, it may be that this overall PEAQS score is too coarse to allow for perceptions of the aesthetic qualities of the field site to emerge as a significant correlate of state body appreciation change scores. Overall, however, our results suggest that dispositional traits—particularly self-compassion—may exert a stronger influence on state body changes in natural environments than environmental factors.

A number of limitations and issues may have affected our findings and their generalisability. One of these concerns is related to the method of recruitment; it is possible that those who agreed to participate in our research differed from the people who declined to participate, e.g., in terms of the dispositional traits measured here or in unmeasured traits. Relatedly, our findings are limited to (predominantly white) women, although it should be noted that previous pre–post studies have reported equivalent results among women and men [11,19,20]. More problematically, because we were reliant on a college sample, we cannot be certain that our results will be generalisable to all population segments. Although there is now a growing body of evidence that suggest that the effects of in situ nature exposure on state body appreciation are robust across diverse national contexts [11,19,20], it may still be useful to replicate the present findings in more diverse cultural and social identity groups.

Additionally, because of a lack of validated measures for use in the Polish context, we were forced to assess the factorial validity of some of our measures in the present study. In some cases, we were able to retain full sets of items for analysis, although this was not the case with the CNS and the PEAQS. Although we followed best practice guidelines in determining the dimensionality of scores on these instruments [49], it should be noted that our sample size was relatively small. As such, the findings of the present study *vis-à-vis* factorial validity should be considered preliminary and requiring replication. In a similar vein, to minimise participant burden, we only measured a small set of dispositional and environmental factors that may have affected the results. Future work could extend this aspect of our design by including additional measures, such as mood, affect, and previous experiences and/or contact with natural environments. An alternative strategy would be to use an experience sampling methodology, wherein participants are asked to report on their state body image at multiple time-points during a walk [15,68]. This would allow scholars to better understand when positive change in terms of body image outcomes begins and peaks.

## 5. Conclusions

These limitations aside, the present study adds to research showing that exposure to natural environments produces significant improvements to state body appreciation and suggests that dispositional factors may be associated with body appreciation outcomes in natural environments. Of particular importance is that we were able to demonstrate this effect in wintry conditions, in a group setting, and in a hitherto neglected national setting, which suggests that these effects may be relatively robust. These results have important practical implications; to the extent that short-term improvements in state body appreciation can be translated into longer-term elevations to trait body appreciation, natural environments may offer an effective means of promoting healthier body image and attendant downstream outcomes, including healthier psychological well-being. More generally, the present findings highlight the importance of ensuring that populations have access to restorative natural environments, which may be a cost-effective means of promoting healthier body image.

## Figures and Tables

**Figure 1 ijerph-19-14548-f001:**
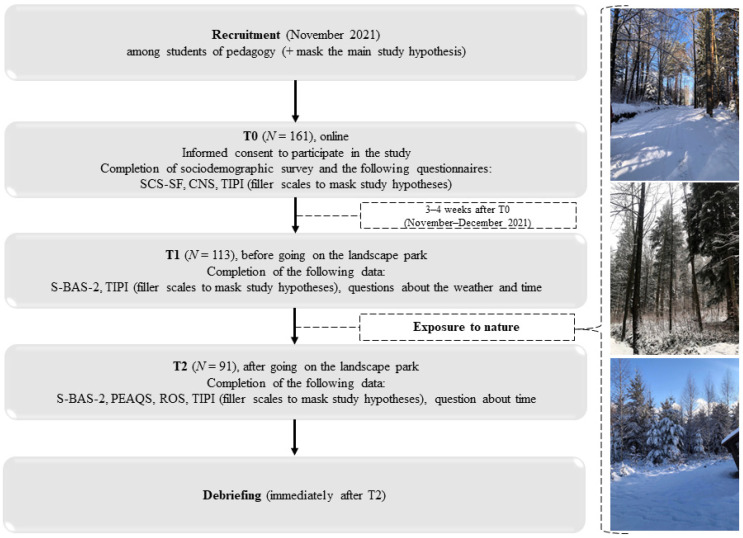
Flow diagram of participants and measurements throughout the study.

**Table 1 ijerph-19-14548-t001:** Descriptive statistics and Pearson’s correlation coefficients.

	1	2	3	4	5
1. Body appreciate change					
2. Connectedness to nature	0.12				
3. Self-compassion	0.22 *	0.38 **			
4. Perceived aesthetic qualities	0.12	0.28 *	0.21 *		
5. Subjective restoration	0.18	0.30 *	0.25 *	0.61 **	
*M*	3.56	3.47	5.32	5.75	3.90
*SD*	0.79	0.87	1.54	1.28	0.88

* *p* < 0.05, ** *p* < 0.001.

**Table 2 ijerph-19-14548-t002:** Prediction of post-walk state body appreciation.

		State Body Appreciation Change				
Step	Variables	*B*	*SE*	β	*t*	*p*
1	Connectedness to nature	0.22	0.19	0.24	2.17	0.033
	Self-compassion	0.28	0.10	0.31	2.80	0.006
2	Connectedness to nature	0.17	0.10	0.18	1.66	0.100
	Self-compassion	0.32	0.10	0.35	3.08	0.003
	Perceived aesthetic qualities	0.01	0.06	0.03	0.21	0.837
	Subjective restoration	0.07	0.05	0.19	1.42	0.159

## Data Availability

All data are available from the corresponding author.

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
