# Peer review of "The Impact of a Woodland Walk on Body Image: A Field Experiment and an Assessment of Dispositional and Environmental Determinants"

_ijerph, 2022, doi:10.3390/ijerph192114548_

Round 1

Reviewer 1 Report

Well written manuscript. Only a couple of typos to correct. e.g. line 389-390: ''...that the those...'' should be ''...that those...''

In the discussion, the discrepancies of the experimental conditions with those from the literature should be discussed. As an example, in your experimental conditions, here was snow unlike in the rest of the literture. Is the presence of snow in your study may have an effects on mood  state ? Snow reflects the light and increase lighting and could have an effect on mood and decrease depressive states. It would be interesting to discuss it with ressources from the literature (e.g. Leibowitz, K., & Vittersø, J. (2020). Winter is coming: Wintertime mindset and wellbeing in Norway. International Journal of Wellbeing10(4).)

Author Response

Thank you for your email to us concerning our manuscript with the title above, which we submitted for your consideration at the International Journal of Environmental Research and Public Health. Thank you also for the opportunity to submit a revision of our manuscript for your further consideration. We have now had an opportunity to consider the comments from the two reviewers, which we found very helpful. Based on those comments, we have prepared a meticulous revision that responds to each of the suggestions we received. We describe the revisions that were made to our manuscript in more detail below. To facilitate this process, our responses are placed below the comments (in bold script) from the reviewers and any changes to the manuscript have been tracked using the Track Changes in MS Word. We hope you agree that our manuscript is closer to fulfilling its publication potential. However, if there are any further issues requiring our attention, we would be grateful for a further opportunity to work together with you and the reviewers toward rectifying those issues.

Reviewer 2 Report

While I think your sample is limited, as you indicated, by the majority of white, college educated women, I find the study very interesting and your analysis of the sample very good.  I think your study was robust in the measures you chose.  

I would have liked to have seen you more clearly define the field measures.  How many walks were taken during the experimental phase.  You define where the walks took place and their average length of time, but it die not indicate how many walks.  Also I would have liked to have seen more written about when positive change began or if you only assessed change after all of the walks were complete

Author Response

(The authors gave the same response as above.)
